# Children's risk preferences vary across sexes, social contexts, and cultures
Roman Stengelin [1,2] ✉, Luke Maurits[1], Robert Hepach[3] & Daniel Benjamin Moritz Haun [1]

People exhibit more risk-prone behaviors when together with peers than when in private. The interplay of social context effects and other variables that alter human risk preferences (i.e., age, sex, or culture) remains poorly understood. Here, we explored risk preferences among Namibian Hai‖om and Ovambo children ($N$ = 144; $Age_{Range}$ = 6–10 years). Participants chose between risky and safe options in private or during peer presence. In a third condition, children collaborated with peers before their risk preferences were assessed in those peers' presence. Children from both societies were risk-averse, but Hai‖om children showed greater risk aversion than their Ovambo counterparts. Across cultures and ages, boys were less averse to risks than girls. This effect was most pronounced during peer presence, whereas collaboration did not additionally affect risk preferences. These results suggest a dynamic interplay of individual, social, and cultural factors shaping children's risk preferences.

From foraging strategies among hunter-gatherers to stockbrokers' financial investments – navigating situations with uncertain outcomes is an integral part of human economic decision-making. Weighing safer (little to no variation in possible outcomes) and riskier options (significant variation therein), rational agents should choose options with highest average payoffs, irrespective of variation in outcomes. Research in economics and psychology suggests that humans do not always act rationally: Humans exhibit variant risk preferences, ranging from risk proneness to risk aversion, based on a variety factors, including age, sex, and culture[1–4], and the social context of the decision.

Social context is considered "a crucial evolutionarily relevant context"[5] to the study of human risk preferences and decision-making more generally. Individuals seek more risks when observed by peers as compared to private contexts[6–9]. At the same time, adults' risk preferences do not budge in response to others' economic risks– it is the mere presence or attendance of other agents that promotes risk-seeking behaviors[10]. Risk preferences are shaped by social comparisons[11] and assumed to serve a signaling or reputational function, such as to display strength or competitiveness to peers[5,12–14].

Other social contexts in which humans may encounter economic risks are those in which individuals attain resources interdependently through collaboration[15–17]. Following De Petrillo and Rosati[5], "humans exhibit both extensive and flexible cooperation compared with other primates, and this dependence on social exchange strategies may have shaped our responses to risk." (p.7).

Already children rely on collaboration in various contexts to access activities and resources they could not attain individually[17,18]. Collaborative

contexts are known to alter children's behavior, such as their prosociality[19–23], trust[22], willingness to delay gratification[24], and task motivation[25,26]. Given the significance attributed to collaboration in human evolution, it is important to scientifically test the potential impact of collaboration on risk preferences beyond potential effects of peer presence.

More generally, most research investigating social context effects on risk preferences focused on adolescents and adults from affluent, urban communities in the industrialized Global North[6–9]. The developmental roots of such effects in childhood and their generalizability beyond such particular cultural milieus remain speculative. Prior research highlighted human risk preferences as a developmentally dynamic phenomenon. Adults are less prone to risks than children and adolescents[27–33]. Interestingly, developmental variation in risk preferences is not limited to humans: Chimpanzee risk-taking peaks in young adulthood and decreases later in life, indicating shared developmental dynamics in risk preferences across human and non-human primates[34].

Additionally to this developmental variation, human risk preferences also vary across sexes[3,35–38]. Overall, women are considered more averse to risk than men, and these differences already emerge in middle to late childhood[27,28,39–42]. Such effects may be rooted in distinct behavioral patterns across the sexes[5,43–47]. Support for this notion comes from Apicella and colleagues[43], who assessed risk preferences in a Hadza hunter-gatherer community in Tanzania. Like many other contemporary hunter-gatherer societies, the Hadza value egalitarianism as a fundamental cultural scheme, which includes an appreciation of sex equality[48,49]. At the same time, hunter-gatherer societies often divide subsistence activities by sex[50] such that the outcomes of men's subsistence activities are considered more variable than

[1]Department of Comparative Cultural Psychology, Max Planck Institute for Evolutionary Anthropology, Leipzig, Germany. [2]Department of Psychology and Social Work, University of Namibia, Windhoek, Namibia. [3]Department of Experimental Psychology, University of Oxford, Oxford, United Kingdom. ✉e-mail: roman_stengelin@eva.mpg.de

those of women[43]. The authors found Hadza men to be more prone to economic risks than women, a pattern that emerged robustly across multiple indicators of risk preferences. Interestingly, sex-level variation in risk preferences was already evident in mid-to-late childhood, as boys sought economic risks more than girls. The authors concluded that gender variation in risk preferences might reflect an "evolutionarily selected, species-typical patterns of behavior" (p. 601). Again, comparative evidence supports this interpretation, with chimpanzee males exhibiting higher risk preferences than females[34]. If so, developmental and sex-level effects on risk preferences should be relatively invariant across human populations.

Given psychology's reliance on sampling participants overwhelmingly from urban communities in the North America, Europe, and Australia[51,52], cross-cultural research outside these communities is much needed to assess and increase the generalizability of data and theory. Indeed, cross-cultural research has revealed universal and culturally-specific aspects of human risk preferences[4,53]. For example, adults' risk preferences vary cross-culturally with different subsistence strategies[43,53,54] and, relatedly, as a function of economic wealth and socioeconomic variables[4,55,56]. The common focus on the most wealthy and industrialized communities of the Global North is thus poorly suited to investigate human risk preferences more generally.

Cultural variation in risk preferences already emerges in mid-to-late childhood[57]. In this study, children from India, the U.S., and Argentina exhibited higher risk preferences than Ecuadorian Shuar children. This effect was attributed to differing degrees of market integration in these communities, such that higher risk preferences coincided with market integration. Indeed, Shuar children from a more market-integrated community showed greater risk preferences than their peers from more remote, subsistence-based communities[57]. While this research did not report cultural differences in sex disparities for children's risk preferences, it's important to note that the study design did not directly test for such interactions. Instead, sex was modeled as a fixed effect, invariant across cultures.

Cárdenas and colleagues[58] studied 9- to 12-year-old Colombian and Swedish children with regard to their risk preferences. Boys exhibited more risk-prone attitudes in both societies than girls, but this effect was more pronounced among Colombian children. Crucially, this pattern aligns with the more egalitarian gender norms in Sweden compared to Colombia, suggesting that different risk preferences across the sexes result partly from cultural learning[58]. More support for this view comes from a study by Liu and Zuo[59], who assessed risk preferences among two Chinese communities, the matrilineal Mosuo and the patriarchal Han. Around the age of 9 years, Han girls were more averse to economic risks than boys, reflecting sex differences observed in previous work. However, a different pattern emerged among same-aged Mosuo children: Boys were more risk-averse than girls, but this effect reversed later when children from both communities attended schools together. According to the authors, this finding reflects distinct gender roles among the Mosuo where women predominantly make economic decisions.

In summary, the more risk-prone attitudes among boys compared to girls in most studies may result from similar gender roles prevalent in the researched communities: Industrialized, urban societies of the Global North[39,41] and, to some extent, rural hunter-gatherer communities in the Global South[43] both enculturate boys to engage in risky economic decision-making. Cultural learning and evolutionary predispositions[34,43] work in tandem to shape human risk preferences. Middle childhood is a crucial period as children attune to social and cultural norms when navigating economic risks, with subsistence and market integration being significant sources of variation.

While most empirical studies focus on risk preferences of sole decision-makers, real-world economic choices often occur in social contexts, where reputational concerns and interdependence may profoundly impact decision-making. Despite the potent impact of social contexts on human risk preferences, surprisingly little is known about how such effects are shaped developmentally and how they vary cross-culturally. Some research indicates that peer presence effects are more pronounced among adolescents than adults[6,8,60], which suggests developmental variation at later ages. Other research shows that sex differences in risk preferences are more accentuated during peer presence[61,62], but little is known about the developmental roots of such social context effects in childhood. Interestingly, German preschoolers are sensitive to peer presence and collaboration in the context of sharing[20,23,63–65], indicating that social context effects on risk preferences might emerge at this age, too.

Furthermore, prior research studying social context effects on risk preferences almost exclusively sampled urban, industrialized, and wealthy communities in the Global North, seriously limiting the generalizability of such research beyond urban, industrialized, and wealthy communities. A relevant dimension in this regard concerns societal emphases on psychological autonomy compared to hierarchical relatedness[66,67]. Societies in which individuals are equipped with high levels of psychological autonomy, navigating social interactions emphasizing their subjective desires and interests, may be less susceptible to social contexts as decisions can be made without strong social obligations. Given the role of autonomy as a foundational schema among many hunter-gatherer societies[68,69], studying peer presence effects on risk preferences among hunter-gatherer communities offers a critical test of the robustness of such effects. In contrast, individuals from societies emphasizing hierarchical relatedness may be more susceptible to social context when navigating economic risks, given their dense networks of social relations and adherence to social norms[70].

To gain a richer understanding of how social environments shape children's risk preferences, research must encompass cultural communities exhibiting variation in subsistence, but also in socialization goals regarding child autonomy and interdependence. Such perspectives are critical for improving the generalizability and validity of developmental research across psychology and economics.

The current study investigated the development of children's risk preference in social context. By manipulating the presence of and interdependence with peers in children's economic decision-making, we explored how age, sex, and cultural background affect how social interactions shape their risk-taking behavior. We focused on 6- to 10-year-old children as a crucial phase in the emergence of culturally-specific risk preferences[57–59]. To move beyond frequently studied participant pools from urban and industrialized communities in the Global North, we studied children from two neighboring communities in rural Namibia, the Ovambo and the Hai‖om. Across multiple trials, children were tasked to move two containers down a ramp. We assessed boys' and girls' risk preferences in same-sex dyads to test social context effects on risk preferences across the sexes[61,62].

Across trials, a transparent container (i.e., a hollow ball) was baited with one reward (*safe option*) while the second opaque container was loaded with either two or none (*risky option*). Children could see the two types of containers but, crucially, could not know the content of the risky option container. They could either choose the risky or the safe option in each trial. Children engaged in three experimental conditions manipulating the social context of their risky decision-making, each assessed within subjects. In the *individual condition*, they chose privately (i.e., no peer present). In the *observation condition*, children were observed by a peer during the decision process. In the *individual* and the *observation* conditions, children worked independently when moving the ball down the ramp and choosing the risky or safe options. In contrast, in the *collaboration condition*, children collaborated with a peer when moving the ball down the ramp but then made their decision independently.

We assessed children from two culturally diverse but geographically close communities: the psychologically-autonomous Hai‖om and the hierarchically-related Ovambo. This approach allowed us to mitigate potential issues in cross-cultural work on risk preferences, such as by confounding country-level factors (i.e., economic wealth or political system[4]) or variation in ecology and climate[71] with other aspects of cultural experience. While the Hai‖om and Ovambo reside in similar physical environments, they diverge starkly in traditional subsistence practices and in cultural emphases on (psychological) autonomy compared to (hierarchical)

relatedness. The Hai||om have traditionally relied on hunting and gathering as subsistence. Due to colonialization and political marginalization, their lifestyle significantly changed over the last decades, Still, many aspects of their traditional subsistence and cultural schemas remain integral to their cultural identity. The Hai||om communities assessed in the current study mostly practice a mixed economy combining foraging (gathering nuts, berries, and tubers, whereas hunting big game is not permitted by law) with other forms of subsistence (i.e., governmental pensions, cultivating crops). The Hai||om raise children to develop high levels of psychological autonomy and instill an egalitarian ethos[72]. These emphases are foundational schemas among foraging societies[68,73]. While ethnographers have highlighted that the Hai||om exhibit patrilineal organization of community life (transmission of communal headman-ship and naming[74]) and gender-segregated division of labor, the egalitarian ethos among the Hai||om renders strict categorizations of them as patrilineal unduly rigid[75].

Compared to the Hai||om, psychological autonomy and egalitarianism are less central among the Ovambo. Instead, the Ovambo value hierarchical relatedness as an important socialization goal – as is prototypical for rural, subsistence-based farming ecologies[70]. The traditional subsistence system of the Ovambo is that of farming (i.e., millet, beans) and herding (i.e., cattle, goats). Their agropastoral lifestyle remains integral to the cultural identity of many Ovambo families in rural areas. Many rural Ovambo families practice a mixed-subsistence lifestyle incorporating both wage labor and agropastoralism organized within the extended family. In contrast to the Hai||om, the Ovambo kinship system is structured matrilineally[76]. They practice a strongly segregated division of labor across the sexes, with men often responsible for income and economic decisions.

Following previous work, we expected younger children to be more risk-seeking than older participants[29,30,32]. Furthermore, we hypothesized that girls from both societies would be more risk-averse than boys[3,41,43] but assumed that this effect could be more pronounced among the hierarchically-related and gender-segregated Ovambo compared to the more egalitarian and autonomous Hai||om. Across sites, we expected that children would make more risky decisions in the presence (i.e., observation, collaboration) compared to the absence of peers (i.e., individual)[6–9]. Given the lack of evidence on the potential effects of collaboration on risk preferences, we were agnostic as to whether the respective condition would affect children's risk preferences above and beyond the effect of peer presence in the observation condition. Given the cultural emphasis on psychological autonomy among the Hai||om, we assumed that the potential effect of condition would be weaker among these children compared to the relational Ovambo.

To test the interplay of individual-level and social-contextual variation in risk preferences, we further tested if sex differences in risk preferences would be more pronounced during peer presence[61,62]. Assuming the reputational function of risk preferences aligns with gender norms, we expected boys' decision-making to be more sensitive to peer presence than girls'. If so, we expected such an interaction to be particularly pronounced among the hierarchically-related Ovambo and to a lesser extent among the egalitarian and autonomous Hai||om. Finally, we assumed that effects of culture, condition, and sex would become more pronounced with age in response to an increasing internalization of gender-segregated norms and activities. Notably, we left these hypotheses unspecified in our preregistration, which merely communicated the procedure of this investigation. Also, our statistical approach deviates from our initial strategy (see below). We thus emphasize that these hypotheses should be treated exploratory, rather than confirmatory.

## Methods

### Participants
We tested 72 Hai||om children ($M_{Age}$ = 8.09 years, range = 6.41–10.11, 36 girls, 36 boys) and 72 Ovambo children ($M_{Age}$ = 8.20 years, range = 6.27–10.25, 36 girls, 36 boys). Information on children's sex was obtained from their birth certificates and caregivers, while data on children's ethnicity was gathered from school teachers and caregivers. Children could

also partake as a confederate peer in the observation and collaboration condition with other participants following their participation in the study. Four Hai||om children (three girls) and three Ovambo children (one girl) acted as confederate peers in the study before they had participated in the study themselves (see Supplementary Note 6 for additional analyses). Given the research was conducted in rural communities, all participants and confederates were familiar with each other. Participants and confederate peers were paired randomly from all same-aged children (±2 years) available.

Hai||om participants were recruited and tested in two villages, 31 children from a village comprised of approximately 300 inhabitants and roughly 50 km from the nearest town and 41 children from a community of around 800 inhabitants situated about 40 km from the nearest town. Social and family ties between both communities are substantial such that people often move flexibly between the two communities to visit friends and relatives. We also observed temporary moves of households between communities during the time our research took place. For this purpose, people typically travel the neighboring bush- and farmlands by foot, which is about a day's walk. Finally, Ovambo children from a small town with around 4000 inhabitants participated in the study.

Children were tested at local school buildings by a male experimenter well-known to these children from previous research projects[26,77,78]. Parental consent (recorded via videotape or written consent, depending on parental literacy and preferences) and written consent of school principals were obtained before the study. Children's verbal assent was obtained before each study session.

### Ethics and inclusion statement
The Ethics Committee of Leipzig University approved the research. Furthermore, we gained approval from the Ministry of Home Affairs and Migration of the Republic of Namibia, the Regional Council of Education in Namibia, and the Working Group of Indigenous Minorities in Southern Africa (WIMSA). Further, results were presented and discussed at a public lecture at the University of Namibia in 2022. Local research assistants were included in the translation of study instructions and the creation of video instructions as part of an ongoing work contract. The current research is part of a long-term research collaboration between the first author and the communities hosting the research[79].

### Design and materials
Children engaged in three conditions (individual, observation, collaboration) in a within-subjects design. Each condition comprised four trials that were assessed block-wise, amounting to 12 trials per participant. We counterbalanced the order in which conditions were presented across participants.

We utilized a wooden device (length = 70 cm, width = 20 cm, height = 18 cm) to assess children's risky decision-making. A transparent cover enabled children's visual access to the inner parts of the apparatus while blocking physical access to it. The experimenter placed two hollow plastic balls (diameter = 3 cm) on the upper side of a wooden ramp in each trial. Transparent balls containing one piece of candy (bought in local shops) served as *safe options.* The content of these containers was evident throughout the study. Blue (opaque) balls reflected *risky options* as they contained either two pieces of candy or none (each with a probability of $p = 0.5$). Here, the content of the balls could not be reliably estimated before opening them, such that participants, confederates, and the experimenter did not know whether a risky option would contain candy or not (see Supplementary Note 1 for depiction of study materials). To ensure that neither children nor the experimenter could infer the risky option's content, we filled the empty balls with gravel of similar sizes and weights to the candy rewards before the study.

We aligned five wooden barriers on the wooden ramp. These barriers blocked the balls from rolling down the ramp unless participants lifted them by pulling ropes attached to the barriers. Once both balls (risky and safe options) had rolled down the ramp, participants could sit on a cushion at the

Fig. 1 | **Illustration of study protocol across conditions.** **a** Individual condition; **b** observation condition with peer present; **c** collaboration condition with peer present; Illustration by Leonore Blume, used with permission.

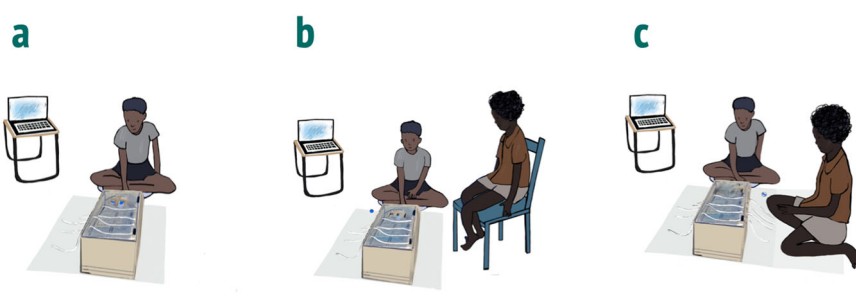

Fig. 2 | **Retrieval of risky or safe option.** **a** Children's choice of either safe reward (transparent ball on the left-hand side) or risky reward (blue opaque ball on the right-hand side); **b** Risky choice; **c** Safe choice; Illustration by Leonore Blume, used with permission.

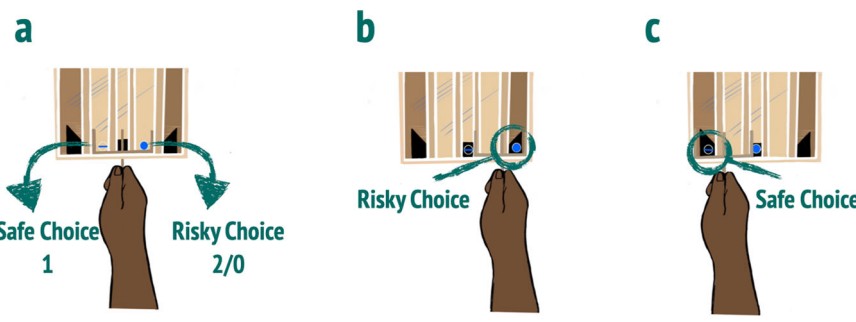

respective end of the device to choose their preferred option. To do so, they moved a wooden handle either to their right- or left-hand side. Participants had to move the handle to their left to select the left ball, after which the chosen option fell out of the apparatus. If so, the remaining (right) ball slipped out of reach inside the device (and vice versa). We counterbalanced the positions of balls across participants and conditions. The risky option was twice located on their left-hand side and twice on the right-hand side for each child and condition.

Children could store each ball obtained on a red tray. These balls were only opened after participants completed all 12 trials to ensure that children would not update their preferences from luck or misfortune hitherto. Instructions were given via audio and video files embedded in a presentation (Microsoft PowerPoint) played on a laptop. Instructions were translated from English by two independent research assistants. We resolved disagreements between coders through discussion between translators and the first author.

## Procedure

The experimenter led the target child into the testing room while their confederate waited outside until they could join the study. The experimenter asked the target child to sit next to the apparatus and attend to the instructions presented on the screen.

On-screen, the risky and safe options were introduced to the target child. For this purpose, four transparent balls arranged on a tray were shown on screen. Next, the content of the balls was shown (one candy per ball) and children were briefed that every transparent ball contained one candy. To solidify children's understanding of these options, the experimenter presented the target child with a similar tray on which four transparent balls were arranged. He opened these balls one after another and highlighted that each transparent ball contained one piece of candy ("The transparent balls always contain one candy. You see? In every transparent ball, there is exactly one candy.").

The next video depicted a set of four opaque balls. Instruction was given similarly to the transparent balls, except that it was unclear whether a blue ball would contain two candies or none. Again, the content of the balls was revealed, confirming this information. The experimenter presented the target child with a tray comprising four blue balls. He opened them together with the children and emphasized that they contained either two pieces of

candy (two balls) or no candy (two balls; "The blue balls are risky. Sometimes they contain two candies. Sometimes, they do not contain any candy, but only a stone. Do you see? With the blue balls, you never know: Maybe you get even two candies, or maybe you do not get any candy.").

The following video instructed children how to choose between risky and safe options in a given trial. Children observed four separate videos in which an agent first moved the handle to the left to obtain a blue ball, then moved it to the right to receive a transparent ball, followed by moving the handle to the left to receive a transparent ball, and finally moving it to the right to receive a blue ball. Next, the experimenter modeled the retrieval of a transparent and a blue ball on either side before the target child could practice themselves in all possible ways. The experimenter started the test phase after the target child successfully moved the handle to their left- and right-hand sides for each option.

Next, the test phase started with the order of conditions being counterbalanced across children. In the individual condition, only the experimenter was present during these trials out of sight of the target child turning his back to them. In the observation condition, the confederate peer entered the room and sat next to the target child on a chair. The confederate was neither involved in the target child's effort to lift the barriers nor in their subsequent choice between risky or safe options. In the collaboration condition, the confederate was also present. Here, the target child and their confederate collaboratively pulled the ropes to lift the barriers jointly. Similar to both other conditions, the target child chose their preferred option independently of their confederate (conditions are depicted in Fig. 1).

Before each block, we showed children a video in which one (individual condition; observation condition) or two models (collaboration condition) lifted the barriers by pulling the ropes attached to them one by one. Next, we asked them to choose between the risky and safe options by acting accordingly (see Fig. 2). Participants were reminded of the container-content contingencies at the start of each block (i.e., "Now you can choose which ball you want to have. The transparent, safe one, or the opaque, risky one."). Per block, children could collect four balls. Between blocks, we asked both the target child and their confederate to briefly wait outside the testing room such that the experimenter could prepare the following study phase. After children had collected 12 balls across all three conditions, the target child and the experimenter opened the balls to retrieve the candy obtained

throughout the study. Confederate peers received a fixed amount of two candies to compensate them for their assistance.

## Coding

A coder blind to research hypotheses and uninvolved in the research project coded children's choices from video (main coding). They coded risky choices as '1' and safe choices as '0'. The first author was the reliability coder for the entire sample to assess interrater reliability. Interrater agreement was excellent across conditions (all $ICCs \geq 0.97$). We used the main coding data to conduct statistical analyses.

## Data analyses

Initially, we planned to count children's risky choices separately for each condition and later implement these sum scores as a primary outcome in our statistical analyses. We drafted a preregistration according to this approach (https://osf.io/aqky5). While running the study, we realized that some children's choices were driven by individual-level side biases. This tendency invalidated our preregistered approach of using sum scores to model children's risk preferences[80]. To address this issue, we excluded children with strong side biases (i.e., those who *only* pulled the handle to their left ($n = 17$) or to their right ($n = 2$) since these children had shown no variation in their responses throughout the study. This decision arose when inspecting the data and was thus not preregistered. However, the research design ensured these excluded children encountered the risky option in half the trials due to counterbalancing. Consequently, their inclusion or exclusion would not significantly skew the overall dataset. To confirm this, a separate analysis including all participants yielded consistent results, solidifying the robustness of our findings (see Supplementary Note 4). Further, we decided to model children's risk preferences on a trial level by estimating the probability of a child choosing the risky option while accounting for individual-level side biases. We fitted generalized linear mixed models with Bernoulli response distributions in $R$[81] using *brms*[82,83].

For each trial, we tested the effects of social context (i.e., *condition*: 3 levels [individual; observation; collaboration]), *age* (scaled to a mean of 0 and a standard deviation of 1), *sex* (centered around 0), and *culture* (centered around 0) on children's risk preferences. These variables were added as fixed effects into the model. To assess the interplay of these variables, we also added all corresponding interaction terms (2-way, 3-way, 4-way) to the model. To account for individual-level side biases, we added random slopes of *side* (2 levels [right; left]) per individual.

Rather than arbitrarily assigning one of our three conditions a reference role corresponding to an intercept term and adding two dummy variables for the others, we directly estimated three separate probabilities of risk taking for the three conditions[82]. Also included were fixed effects of sex, culture, age, and the side of the apparatus with the risky option in each trial. Each of these variables was centered around 0 to ease interpretation and prior choice for the three per-condition probabilities. Sex, culture and age were modeled to interact with condition and each other. We used tight (double exponential) priors centered around 0 on all interaction terms to encourage pooling of data in the absence of strong evidence for such interactions. In this way we avoided inflated uncertainty in our estimates of the effects of condition as the main experimentally manipulated variable of interest. For the other parameters we used normal priors centered around 0. We provide posterior mean probabilities of children choosing the risky option together with the associated 95% highest posterior density intervals (HPD) for each combination of the predictors. We report whether these HPDs include chance level (0.50) to categorize children as risk averse, risk neutral, or risk seeking. Pairwise contrasts of the posterior probabilities for risk preferences are given as indicators of effect size of the categorical predictors condition, sex, and culture, with age set at its mean. To further visualize how the interplay of these variables unfolds developmentally, we also plot the posterior probabilities as estimated by the full model. As preregistered, we additionally assessed children's risk preferences in the first condition they had engaged in to eliminate carry-over effects due to the within-subjects research design. The results of this analysis corroborate

those of the main data analyses and are thus reported in the Supplementary Note 2. Further analysis also confirmed stable risk preferences throughout the test phase (see Supplementary Note 5). Data and code are available at https://osf.io/3nukt/ (see also Supplementary Data 1 and Supplementary Note 3 for the Codebook).

## Reporting summary

Further information on research design is available in the Nature Portfolio Reporting Summary linked to this article.

## Results

Children of both sexes and across cultures preferred the safe over the risky option. This risk aversion was particularly pronounced in the individual condition for both Hai‖om (probability_{Boys} [95%-HPD] = 0.28 [0.18; 0.39], probability_{Girls} [95%-HPD] = 0.20 [0.12; 0.30]) and Ovambo children (probability_{Boys} [95%-HPD] = 0.32 [0.22; 0.44], probability_{Girls} [95%-HPD] = 0.27 [0.18; 0.38]). In the presence of their peers, children showed a somewhat more risk prone attitude. However, most children tested in the observation condition (Hai‖om: probability_{Boys} [95%-HPD] = 0.49 [0.26; 0.52], probability_{Girls} [95%-HPD] = 0.22 [0.12; 0.33]; Ovambo: probability_{Boys} [95%-HPD] = 0.49 [0.26; 0.52], probability_{Girls} [95%-HPD] = 0.31 [0.20; 0.43]) and the collaboration condition (Hai‖om: probability_{Boys} [95%-HPD] = 0.33 [0.21; 0.46], probability_{Girls} [95%-HPD] = 0.16 [0.08; 0.46]; Ovambo: probability_{Boys} [95%-HPD] = 0.42 [0.29; 0.56], probability_{Girls} [95%-HPD] = 0.32 [0.21; 0.44]) remained averse to risks (see Supplementary Note 7 for all model parameters).

Pairwise contrasts revealed that, across cultures, boys were more prone to risks when together with peers than when being alone (Observation > Individual: contrast_{Hai‖om} = 0.98, contrast_{Ovambo} = 0.88; Collaboration > Individual: contrast_{Hai‖om} = 0.81, contrast_{Ovambo} = 0.96; note that contrasts of 0.50 indicate no effect of condition). There was no evidence supporting claims on collaboration fostering risk preferences beyond the effect of peer presence among boys (Collaboration > Observation: contrast_{Hai‖om} = 0.17, contrast_{Ovambo} = 0.71). Ovambo girls showed a less pronounced trend toward higher risk preferences during peer presence (Observation > Individual: contrast_{Ovambo} = 0.77; Collaboration > Individual: contrast_{Ovambo} = 0.82) and no clear evidence for an effect of collaboration beyond the effect of peer presence (Collaboration > Observation: contrast_{Ovambo} = 0.58). Finally, there was no clear evidence that Hai‖om girls' risk preferences were subject to peer presence (Observation > Individual: contrast_{Hai‖om} = 0.63; Collaboration > Individual: contrast_{Hai‖om} = 0.16; Collaboration > Observation: contrast_{Hai‖om} = 0.11). Note that these contrasts indicate that Hai‖om children of both sexes appeared even more averse to risk following collaboration compared to individual settings or mere peer presence, whereas Ovambo children did not exhibit such a tendency.

We found robust sex differences in children's risk preferences across cultures and conditions. Among the Hai‖om, boys were more prone to choose risky options both when in private (Boys > Girls: contrast_{Individual} = 0.90) and during peer presence (contrast_{Observation} = 0.99; contrast_{Collaboration} = 0.99). A similar pattern emerged among the Ovambo in both individual contexts (Boys > Girls: contrast_{Individual} = 0.78) and during peer presence (contrast_{Observation} = 0.83; contrast_{Collaboration} = 0.88).

Overall, Ovambo children exhibited stronger risk preferences than their Hai‖om counterparts. For girls, this effect persisted across all conditions (Ovambo > Hai‖om: contrast_{Individual} = 0.89; contrast_{Observation} = 0.90; contrast_{Collaboration} = 0.99). For boys, a similar effect was evident in the individual (contrast_{Individual} = 0.74) and the collaboration condition (contrast_{Collaboration} = 0.87). For the observation condition, contrasts of posterior predictions indicated no evidence in support of an effect of culture among boys (contrast_{Observation} = 0.49).

Sex differences in children's risk preferences mostly persisted across conditions and cultures, yet these effects were more pronounced during peer presence as compared to individual settings. Hai‖om boys appeared particularly prone to peer presence effects, although a similar tendency was also

**Fig. 3 | Risk Preferences depending on sex, condition, and culture.** Posterior probability distributions illustrating the estimated probability that a child will choose the risky option depending on their sex, condition, and culture. Densities are based on the full model to illustrate all potential interactions between predictors. Children's age is set at the mean. Dotted vertical lines represent chance level at probability$_{Chance}$ = 0.50; $n$ = 125 participants.

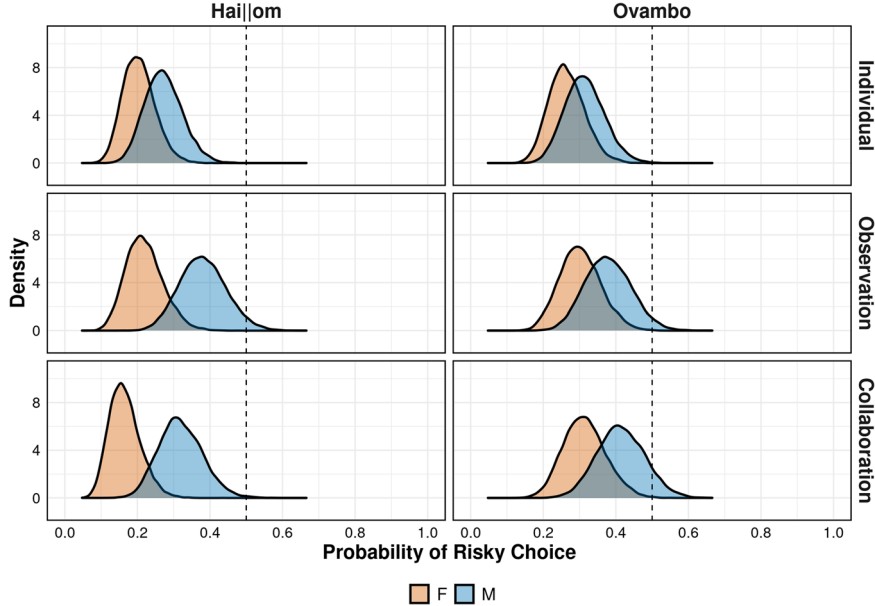

**Fig. 4 | Risk preferences across ages, conditions, and culture.** Posterior probabilities illustrating the estimated probability that a child will choose the risky option depending on their age, condition, and culture. Densities are based on the full model to illustrate all potential interactions between predictors. Children's sex is centered at zero. Solid lines present posterior means, surrounding areas present 95%-HPDs. Dotted horizontal lines represent chance level at probability$_{Chance}$ = 0.50; $n$ = 125 participants.

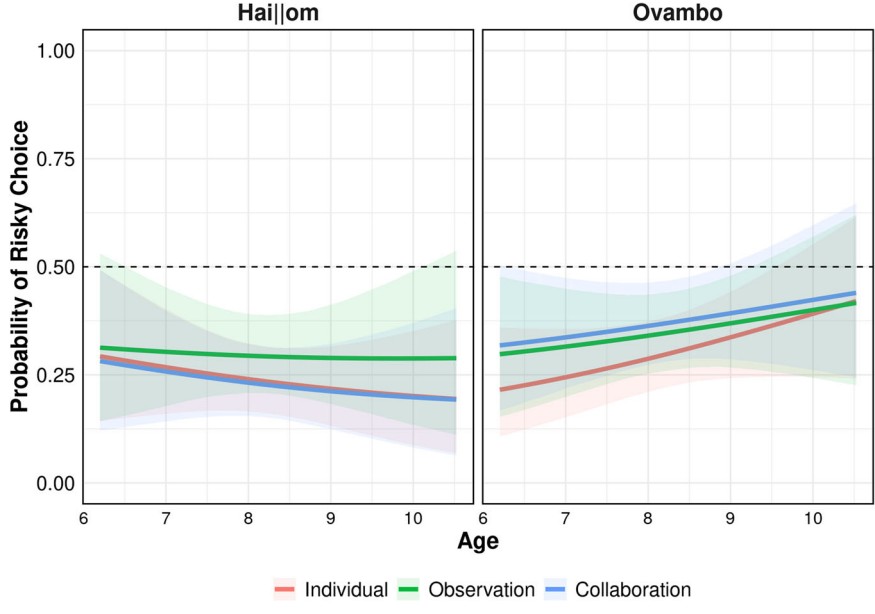

evident among Ovambo boys. Children's risk preferences across social contexts, sexes, and cultures are visualized in Fig. 3.

To further shed light on age-related variation in the effects of condition, sex, and culture on children's risk preferences, we estimated contrasts of the mean posterior probabilities for children's risk preferences at the older and younger age limits tested. For the Hai‖om, results suggested no credible evidence for age-related variation in children's risk preferences. This pattern was found for boys (Older–Younger: contrast$_{Individual}$ = −0.02; contrast$_{Observation}$ = 0.00; contrast$_{Collaboration}$ = −0.09; note that here a contrast of 0.00 indicates stability) and older girls (contrast$_{Individual}$ = −0.14; contrast$_{Observation}$ = −0.04; contrast$_{Collaboration}$ = −0.06). For Ovambo children, estimated contrasts between older and younger children also indicated no credible evidence for age-related variation among boys (contrast$_{Individual}$ = 0.22; contrast$_{Observation}$ = 0.05; contrast$_{Collaboration}$ = 0.10)

or girls (contrast$_{Individual}$ = 0.15; contrast$_{Observation}$ = 0.15; contrast$_{Collaboration}$ = 0.11). Age-related patterns in children's risk preferences are plotted in Fig. 4.

## Discussion

The current study investigated children's risk preferences in three social contexts (individual, peer presence, collaboration) among the Hai‖om and Ovambo of rural Namibia. In contrast to most research conducted among children from urban, Global North communities[30,84], Hai‖om and Ovambo children from rural Namibia were averse to economic risks. On average, risk aversion was more pronounced among the Hai‖om compared to the Ovambo, but children in both communities preferred safe over risky options across all conditions. Across cultures, we found a robust interplay of sex and social context shaping children's risk preferences: Hai‖om and Ovambo boys, rather than girls, made more risky choices when they were accompanied by peers in both social contexts, the observation and the

collaboration condition. In the absence of peers, sex differences in risk preferences were less pronounced. Notably, we found no credible evidence for developmental variation across the age range tested, indicating middle childhood as a period in which effects of sex, peer presence, and culture on children's risk preferences and their dynamic interplay are already being consolidated.

Unlike their urban Global North counterparts[28,30,32], both Hai||om and Ovambo children tested here were, on average, averse to risks. This pattern corroborates previous research with children from rural and subsistence-based communities showing these children exhibit more conservative strategies when navigating economic risks than children growing up in economically wealthy and market-integrated societies[57]. However, it is important to acknowledge that isolating such cultural-level predictors is difficult, as they often co-occur with other meaningful variation in children's socio-cultural experience. Furthermore, risk preferences vary across tasks and reward distributions[5,29], which is why blunt claims regarding children's risk preferences of children in a specific setting and task are problematic unless the adequacy of such generalizations is tested.

Interestingly, we documented risk aversion among children from two culturally-diverse communities within a single geographic region, inhabiting largely similar physical environments. While the Hai||om tested in the current research practice a mixed-subsistence of gathering bushfood with gardening and seasonal wage labor, agropastoralism plays a reduced role among the Ovambo families tested in this study as it is commonly practiced within extended family networks to which children often contribute during school holidays. Previous research has highlighted the advent of agriculture as a key phase in the cultural evolution of human risk preferences[43,53,55]. The current comparison of contemporary hunter-gatherer and agropastoral communities corroborates and adds to a growing body of literature showing that risk-prone attitudes typically observed in industrialized and wealthy societies in the Global North do not easily generalize outside these contexts[4,57].

We found boys, more so than girls, from both cultural contexts were sensitive to peer presence when facing economic risks[61,62]. This adds to a growing body of research highlighting the role of social context in the ontogeny of risk preferences across the sexes among humans and other species[5]. This finding indicates a selective reputational function of risk preferences among boys, which emerges by middle childhood and may persist well into adolescence and later years. Interestingly, the moderating role of sex on children's susceptibility to peer presence is consistent with prior research pointing at similar effects among Dutch adolescents[61,62]. Furthermore, this pattern corroborates theoretical accounts of the reputational function of risk preferences among men as a means to display competitiveness[85].

Collaboration is essential in human economic decision-making[86], but its potential effect on risk preferences remains poorly understood[5]. We did not find an effect of collaboration on risk preferences beyond the effect of peer presence. Among the Hai||om, but not the Ovambo, contrasts indicated a trend toward reduced risk-taking in the collaboration compared to the observation condition. Given the pervasive and early-emerging tendency for collaboration in human children[17,20,26], we explored the effect of collaboration on risk preferences in two diverse cultural settings varying in their emphasis on child autonomy and relatedness. Importantly, children from both communities navigated the collaborative task with ease, as they did in previous research[26]. We propose two alternative explanations to the smaller effect of collaboration on risk preferences among Hai||om children: First, Hai||om's emphasis on children's autonomy as a central socialization goal and cultural schema determines children's collaborative activities in this context. From early on in life Hai||om children are given autonomy to freely engage in social interactions and activities according to their preferences, which may discourage interdependent (i.e., collaborative) modes of action when they are forced upon children (such as in the current study). In line with this explanation, previous work found Hai||om children preferred solitary action over peer collaboration when given the choice between both activities[26]. One may speculate whether the rewarding character of peer collaboration led to a more optimistic bias among the relational Ovambo

compared to autonomous Hai||om children[87–90]. An alternative explanation holds that this effect was driven by increased sharing expectations among Ovambo children engaging in peer collaboration: Peer collaboration evokes fairness expectations in young children, fostering egalitarian sharing with their collaborative partners[19–21]. Among the Hai||om, who practice egalitarian sharing of resources with little emphasis of merit and individual contributions[91], such considerations may have been less critical in the respective condition: These children would employ egalitarian sharing of resources regardless of condition, as we often observe following their research participation. Regardless of which of these explanations holds, these results highlight the need to study risk preferences for resources obtained in interdependent, collaborative settings to gain a more representative understanding of decision-making in naturalistic contexts.

To our surprise, we found no evidence for an interplay between children's risk preferences across sexes, social contexts, and cultures varying across middle childhood. Previous research found children's sensitivity to peer presence peaking in late childhood and adolescence[8]. However, cross-cultural data on risk preferences in the current age range remain limited[43,57]. We aimed to test the developmental origins of individual, social-contextual, and cultural variation in risk preferences at an age at which children's sensitivity to these levels emerges[63,92]. However, the effects we targeted appeared already present at this age. Future research will need to study children below the current age range to shed light on the age at which children become susceptible to gender norms, social contexts, and culture when facing economic decisions involving risk.

## Limitations

The current study assessed children's risk preferences in the domain of gains, rather than losses. Children had the repeated chance to obtain rewards for which they had to invest only minimal effort to gain access to rewards. Previous research has shown that children are more likely to engage in risky decision-making when facing losses, rather than gains[28]. Interestingly, such framing effects appear deeply rooted in human decision-making as they appear across cultures[4] and in other Great Ape species[93,94]. To provide a thorough account of this aspect of human decision-making, further research is required to unravel the developmental interplay of individual-level, social, and cultural variables in children's risk preferences in the domain of losses. This particularly applies to any conclusions drawn from the current study denoting Hai||om and Ovambo children as generally risk-avoidant (see above).

Another limitation of the current study lies in its exploratory character. Our findings align with previous research on cross-cultural and social influences on children's risk preference. Further confirmatory studies are needed to solidify these findings within pre-registered replications recruiting larger samples and diverse populations. Also, such research might establish children's comprehension of the container-content contingencies more explicitly to ensure a better understanding of children's risk preferences in varying social and cultural contexts.

To address our cross-cultural research question and adjust the research to the cultural communities we studied, we developed a new risk preference assessment paradigm. This paradigm incorporates core elements from established risk assessments, such as a forced-choice format with safe and risky options of equal expected values, ensuring face validity[95]. Crucially, it also allows us to investigate children's risk preferences in diverse social settings, examining effects of peer influence and collaboration under controlled conditions. To promote cultural fairness, we used a haptic device to assess children's risk preferences, rather than relying on instructed rules or electronical devices that might induce cultural barriers to the assessment. A key challenge of cross-cultural studies lies in balancing standardized procedures with the unique needs of communities outside the market-integrated and technology-affluent Global North.

It is further important to bear in mind that we installed sunk costs in the procedure, such that children had to invest some effort into moving the reward containers down a ramp before they could choose between them. This approach deviates from most previous work in which children's risk

preferences were assessed for windfall gains[43,57–59]. We did so, given that windfall gains markedly deviate from real-world contexts in which resources commonly need to be accessed before they can be consumed. Installing such sunk costs may alter children's risk preferences toward risk avoidance given the increased value ascribed to resources acquired effortfully[96]. As such, we emphasize that broad classifications of individuals or cohorts as risk-averse or risk-seeking are premature unless situational aspects of the study design are considered accordingly[97,98].

## Conclusions
The current research resonates with and extends previous work showing that children's risk preferences vary across sexes, social contexts, and cultures. A complex interplay of these effects unfolds in middle childhood when boys, more so than girls, adjust their risk preferences based on social context. A generalizable and comprehensive science of human psychology and behavior needs to consider the social and cultural dynamics of child development. Economic decision-making is no exception to this: Navigating economic risks is a social and cultural enterprise.

## Data availability
Behavioral and demographic data to reproduce the analyses in the manuscript and the supplementary materials is available on the Open Science Framework (https://osf.io/3nukt/) and attached to this manuscript (Supplementary Data 1.csv; see also Codebook.pdf).

## Code availability
The analysis code is available on the Open Science Framework (https://osf.io/3nukt/) and attached to this manuscript (Analyses.pdf).

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

## Acknowledgements

We thank Sarah Caldwell, Maleen Thiele, and the reviewers for their helpful comments on the manuscript. We are grateful for the support and assistance of Disney Tjizao, Sarah Leisterer-Peoples, Lisa Hupfer, Wilbard Nambundunga, Leonore Blume, and Linus Useb. We further want to thank the Namibian Ministry of Education, the National Commission on Research, Science and Technology (NCRST), and the Working Group of Indigenous Minorities in Southern Africa (WIMSA) for their organizational support and all participating children and their caregivers for their invaluable trust and support. The funders had no role in study design, data collection and analysis, decision to publish or preparation of the manuscript. This research was funded by internal budgets at Leipzig University and the Max-Planck Society (no funding number assigned).

## Author contributions

R.S.: Conceptualization, Data Curation, Formal Analysis, Investigation, Project Administration, Writing – Original Draft. L.M.: Formal Analysis, Writing – Review & Editing. R.H.: Conceptualization, Supervision, Writing – Review & Editing. D.B.M.H.: Conceptualization, Funding Acquisition, Supervision, Writing – Review & Editing. Illustrations for Figs. 1 and 2 were done by Leonore Blume.

## Funding

## Competing interests

The authors declare no competing interests.
