## [Peer Review File · Communications Psychology]

23rd Oct 23

Dear Dr. Stengelin,

Thank you for your patience during the peer-review process. Your manuscript titled "Peer Presence Effects on Risk Preferences in Childhood: Variation Across Development, Sexes, and Cultures" has now been seen by 3 reviewers, whose comments are appended below. You will see that they find your work of some potential interest. However, they have raised quite substantial concerns that must be addressed. In light of these comments, we cannot accept the manuscript for publication, but would be interested in considering a revised version that fully addresses these serious concerns.

We hope you will find the Reviewers' comments useful as you decide how to proceed. Should additional work allow you to address these criticisms, we would be happy to look at a substantially revised manuscript. If you choose to take up this option, please highlight all changes in the manuscript text file, and provide a detailed point-by-point reply to the reviewers.

Editorially, we consider it important that the revised manuscript address the concerns of Reviewer 1 and Reviewer 3 regarding the sensitivity of the results to inclusion and exclusion criteria as well as their request for additional information regarding participants' task comprehension. Please revise the introduction and discussion to clearly situate your study within the context of past literature and theory.

It is important to clearly indicate which elements are exploratory, which elements have been preregistered, and where deviations from preregistration have been made. As Reviewer 3 points out, there are limitations to the theoretical contribution offered by null-results in exploratory contexts, which our decision making takes into account; in the present case, we appreciate the value of data on underrepresented groups, but highlight that the findings should be suitably described as descriptive, rather than a confirmatory test of theory.

If the revision process takes significantly longer than five months, we will be happy to reconsider your paper at a later date, provided it still presents a significant contribution to the literature at that stage.

Please use the following link to submit your revised manuscript, point-by-point response to the Reviewers' comments with a list of your changes to the manuscript text (which should be in a separate document to any cover letter) and any completed checklist:

[link redacted]

Please do not hesitate to contact me if you have any questions or would like to discuss the required revisions further. Thank you for the opportunity to review your work.

Best regards,

Jennifer Bellingtier

Jennifer Bellingtier, PhD

Senior Editor

Communications Psychology

EDITORIAL POLICIES AND FORMATTING

Editorial Policy: Policy requirements (Download the link to your computer as a PDF.)

Furthermore, please align your manuscript with our format requirements, which are summarized on the following checklist:

Communications Psychology formatting checklist

and also in our style and formatting guide Communications Psychology formatting guide .

* **CODE AVAILABILITY:** All Communications Psychology manuscripts must include a section titled "Code Availability" at the end of the methods section. In the event of publication, we require that the custom analysis code supporting your conclusions is made available in a publicly accessible repository; please choose a repository that provides a DOI for the code; the link to the repository and the DOI must be included in the Code Availability statement. Publication as Supplementary Information will not suffice. We ask you to prepare and upload code at this stage, to avoid delays later on in the process.

* **DATA AVAILABILITY:**

All Communications Psychology research manuscripts must include a section titled "Data Availability" at the end of the Methods section or main text (if no Methods). More information on this policy, is available at <http://www.nature.com/authors/policies/data/data-availability-statements-data-citations.pdf>.

At a minimum the Data availability statement must explain how the data can be obtained and whether there are any restrictions on data sharing. Communications Psychology strongly endorses open sharing of data. If you do make your data openly available, please include in the statement:

We recommend submitting the data to discipline-specific, community-recognized repositories, where possible and a list of recommended repositories is provided at <http://www.nature.com/sdata/policies/repositories>.

If a community resource is unavailable, data can be submitted to generalist repositories such as figshare or Dryad Digital Repository. Please provide a unique identifier for the data (for example a DOI or a permanent URL) in the data availability statement, if possible. If the repository does not provide identifiers, we encourage authors to supply the search terms that will return the data. For data that have been obtained from publicly available sources, please provide a URL and the specific data product name in the data availability statement. Data with a DOI should be further cited in the methods reference section.

REVIEWER EXPERTISE:

Reviewer #1 Cross-cultural Psychology, Developmental Psychology

Reviewer #2 Educational psychology

Reviewer #3 Developmental Psychology, Decision Making

Reviewer #1 (Remarks to the Author):

In this paper, the authors examine the role of culture, gender, and social context on the development of risk preferences among Hai||om and Ovambo children in Namibia. Using a task

that allows children to choose between risky and safe options, they find (1) that overall, children tend to be risk-averse, (2) that Hai||om children were more risk-averse than Ovambo children, and (3) that across culture and age, boys were more risk seeking than girls, especially in cases of peer presence. I think this is an interesting paper that expands our understanding of the social determinants of risk behavior, and addresses important questions about individual decision making versus decision making in social contexts. I have a number of points and comments below that can help strengthen the paper and clarify its contributions.

Methodology:

1. Can the authors speak to children's comprehension of the task? Were comprehension questions included in the protocol? A breakdown of comprehension rates and attempts would be helpful.
2. I'm also a bit confused about the exclusion criteria used for the analysis, especially as they do not appear to be pre-registered. Annotations in the analysis script suggest that exhibiting a "clear strategy" in the game merited inclusion, but those that always pulled left or right were excluded. Can the authors perform analyses including those data points for the Supplement as a point of comparison? The annotations also suggest that participants were excluded as they "did [not?] react to risk". So maybe more generally, some further clarification would be helpful about exclusion and inclusion criteria and the time course of those decisions.
3. Speaking of pre-registration, it does not appear that the authors specified directional predictions (e.g. that boys would be more risk-seeking than girls, that one culture would be more risk-seeking than the other, and that younger kids would be more risk-seeking than older kids), rather, the pre-registration seems much more exploratory ("How is children's risky decision making influenced by - social contexts (individual, observed, collaborative)? - culture? - ontogeny? - gender?"). I would ask the others to be a bit more explicit about which predictions stated in the paper were or were not registered a priori.
4. Did children's choices vary across trials?
5. Do we know if the behavior of the children who acted as the peer observers varied significantly from those who did not?

Organization:

6. The introduction could benefit from a slight reorganization (perhaps adding subheadings?) to make the various components of this investigation more clear: so (1) social context, (2) collaboration, (3) the effect of culture / WEIRD vs. non-WEIRD, and (4) development. And I do think each of those, especially the collaboration and social context sections, can be rounded out with more citations such as:
 - a. Warneken, F., Lohse, K., Melis, A. P., & Tomasello, M. (2011). Young children share the spoils after collaboration. *Psychological science*, 22(2), 267-273.

b. Koomen, R., Grueneisen, S., & Herrmann, E. (2020). Children delay gratification for cooperative ends. *Psychological science*, 31(2), 139-148.

7. A few other citations to consider:

a. Reiter, A. M., Suzuki, S., O'Doherty, J. P., Li, S. C., & Eppinger, B. (2019). Risk contagion by peers affects learning and decision-making in adolescents. *Journal of Experimental Psychology: General*, 148(9), 1494.

b. Harvey, T., & Blake, P. R. (2022). Developmental risk sensitivity theory: the effects of socio-economic status on children's risky gain and loss decisions. *Proceedings of the Royal Society B*, 289(1983), 20220712.

c. Rohde, I. M., & Rohde, K. I. (2011). Risk attitudes in a social context. *Journal of Risk and Uncertainty*, 43(3), 205-225.

d. Defoe, I. N., Semon Dubas, J., & Romer, D. (2019). Heightened adolescent risk-taking? Insights from lab studies on age differences in decision-making. *Policy Insights from the Behavioral and Brain Sciences*, 6(1), 56-63.

8. I also think the argument in the introduction can be strengthened a bit; it could be beneficial to set the argument up in a way that outlines more clearly what's missing in the literature, e.g. "the majority of studies examining risk preferences consider an individual decision-maker acting on their own; however, we know that in the real world, decisions often take place in social contexts, and there are additional signaling and reputational functions that shape our risky choices".

Minor Notes:

9. A minor note, but contrary to lines 112-113, in the Amir 2019 paper (citation 40), we did include gender as a covariate in all of our statistical analyses (it's reported in our model outputs), and it was never a significant predictor.

10. A minor clarification but "we expected younger children to exhibit stronger risk preferences than older participants" (line 213-14) doesn't specify the order of the prediction. I would update to something like "more risk-tolerance" or "more risk-seeking", or even "preferences for risk" instead of "risk preference".

Dorsa Amir

UC Berkeley

Reviewer #2 (Remarks to the Author):

The paper presents a research findings confirming the interplay of cultural and social contexts among children fro two African rural communities. The results reveal a dynamic interplay of developmental, individual, social, and cultural factors shaping children's risk preferences. These findings are are of interest to scholars within the African context as well as the wider public because there is a scarcity of empirical research on African children. Most research in the field is mainly drawn from the Global North which may not be contextually applicable to children from rural communities.

The paper is well written and organized except for very minor omission of closing bracketed content in a few places which can easily be corrected by the author by editing the paper.

The methodology is clearly explained, the sapling and participant sampling and assignment to groups is clearly described, the testing tools and procedures are clearly presented and the data analysis methods and results are clearly and precisely presented and can be replicated.

Reviewer #3 (Remarks to the Author):

This is a very interesting study on the difference in risk preference in children from two culturally-diverse

communities within a single geographic region. The task it self is mostly non-verbal and age appropriate, and the analyses are, although deviating from the pre-registration, well-motivated and sound. In sum there is very little to comment on the task and the analyses.

However, it is not entirely clear what the scope of these results of comparing these to groups of kids really is, in terms of understanding the development of risk preferences or cultural differences.

In the introduction it becomes clear that risk preference vary between ages, sexes, cultures etc. But in the final cultural part specifically no clear framework or predictions emerge. So in the end we also do not know if these results reject or inform a theory about how culture impacts risk preference development. Of course the resulting difference between the two groups are also basically zero, but whether or not that is an issue for an existing theory I would not know. As it currently is I would therefore not see how it would fit the current journal.

This is also related to a lack of information about the generalizability of the current risk task. How well does it relate to other behavioral tasks? Risk preferences are dependent on so many factors, probability, the presence of a safe choice etc. So it is also hard to compare these results to other studies. (ps. see Defoe et al., for good meta-analyses on developmental patterns, they could be a nice addition for a more updated reference list).

In sum, it is currently hard to see how these results fit into the larger scheme of things, in relation to both development of risk preferences and cultural comparisons.

minor. I wondered if it would be interesting to look at order effects.. maybe all risks were taken in the first block? or later when a bunch of safe rewards were harvested?

Also it would be good to explain how the left/right bias was still a problem after removing so many subjects. And how you are really sure that a strong left or right bias represent misunderstanding the task and not reflects a real preference.

Finally, want to note that the paper was very well written and easy to read!

Reviewer 1:

(1)

“Can the authors speak to children’s comprehension of the task? Were comprehension questions included in the protocol? A breakdown of comprehension rates and attempts would be helpful.”

To ensure children understood how to access rewards, we implemented a training phase. Children received hands-on experience by moving a handle left and right, retrieving reward containers placed on both sides.

We employed transparent containers for the "safe" option, allowing children to continuously see the contents throughout the experiment. To solidify understanding, we presented both transparent (safe) and opaque (risky) containers visually: 4 on-screen and 4 physically. We also emphasized these contingencies multiple times in our instructions, both during training and in the test phase. We opted not to include comprehension questions to streamline the introduction and to ensure children’s comfort: Rhetorical questions are relatively rare among Owambo and Hai||om outside of more hierarchical classroom activities.

Following your comment, we now provide more detail on the training and test procedure including instructions together with a discussion of this concern. At the same time, we remain confident all children understood the distinctions between safe and risky choices.

“To solidify children's understanding of these options, the experimenter presented the target child with a similar tray on which four transparent balls were arranged. He opened these balls one after another and highlighted that each transparent ball contained one piece of candy ("The transparent balls always contain one candy. You see? In every transparent ball, there is exactly one candy.") [...] He opened them together with the children and emphasized that they contained either two pieces of candy (two balls) or no candy (two balls; "The blue balls are risky. Sometimes they contain two candies. Sometimes, they do not contain any candy, but only a stone. You see? With the blue balls, you never know: Maybe you get even two candies, or maybe you do not get any candy.")” (p. 13)

“Next, we asked children to choose between the risky and safe options by acting accordingly (see Fig. 2). Participants were reminded of the container-content contingencies at the start of each block (i.e., “Now you can choose which ball you want to have. The transparent, safe one, or the opaque, risky one.”).” (p. 15)

“Another limitation of the current study lies in its largely exploratory character. Our findings align with previous research on cross-cultural and social influences on children’s risk preference. Further confirmatory studies are needed to solidify these findings within pre-registered replications recruiting larger samples and diverse populations. Also, such research might establish children’s comprehension of the container-content contingencies more

explicitly to ensure a better understanding of children's risk preferences in varying social and cultural contexts.” (p. 25)

(2)

“I’m also a bit confused about the exclusion criteria used for the analysis, especially as they do not appear to be pre-registered. Annotations in the analysis script suggest that exhibiting a “clear strategy” in the game merited inclusion, but those that always pulled left or right were excluded. Can the authors perform analyses including those data points for the Supplement as a point of comparison? The annotations also suggest that participants were excluded as they “did [not?] react to risk”. So maybe more generally, some further clarification would be helpful about exclusion and inclusion criteria and the time course of those decisions.”

Indeed, these exclusion criteria were not pre-registered as we did not anticipate such response tendencies. Although we had originally planned to analyze data per block, rather than on a trial level, we were concerned by the relatively high number of participants who chose the option to their left or right-hand side exclusively throughout the study—thus showing no response variation. We believe that a sound analytic approach should be prioritized over strictly following preregistered approaches as long as such choices are communicated transparently. Please note that, given our per-block counterbalancing of side, such biases would not lead to more or less risk preferences but only add noise to the data.

At the same time, such consistent side biases were viable strategies, so we agree that further clarification is needed. We present an additional analysis based on the full dataset in the supplementary materials as well as a more detailed discussion of the post-hoc exclusion criteria in accordance with the reviewer’s suggestion. Results showed that our findings are robust to this exclusion criterion.

“To address this issue, we excluded children with strong side biases (i.e., those who only pulled the handle to their left ($n = 17$) or to their right ($n = 2$) since these children had shown no variation in their responses throughout the study. This decision arose when inspecting the data and was thus not preregistered. However, the research design ensured these excluded children encountered the risky option in half the trials due to counterbalancing. Consequently, their inclusion or exclusion would not significantly skew the overall dataset. To confirm this, a separate analysis including all participants yielded consistent results, solidifying the robustness of our findings. (see Supplementary Materials).” (p. 16)

(3)

“Speaking of pre-registration, it does not appear that the authors specified directional predictions (e.g. that boys would be more risk-seeking than girls, that one culture would be more risk-seeking than the other, and that younger kids would be more risk-seeking than older kids), rather, the pre-registration seems much more exploratory (“How is children's risky decision making influenced by - social contexts (individual, observed,

collaborative)? - culture? - ontogeny? - gender?”). I would ask the others to be a bit more explicit about which predictions stated in the paper were or were not registered a priori.”

Thank you for your valuable feedback regarding the preregistration for our study. To address your concerns, we acknowledge the exploratory nature of the research more explicitly and discuss the need for further research to confirm our results. We also present additional analyses together with relevant recent literature to better embed our findings to mitigate the limitations of the preregistration.

“Notably, we left these hypotheses unspecified in our preregistration, and our statistical approach deviates from our initial strategy (see below). These hypotheses should thus be treated exploratory, rather than confirmatory.” (p. 10)

“Our findings align with previous research on cross-cultural and social influences on children’s risk preference. Further confirmatory studies are needed to solidify these findings within pre-registered replications recruiting larger samples and diverse populations. Also, such research might establish children’s comprehension of the container-content contingencies more explicitly to ensure a better understanding of children’s risk preferences in varying social and cultural contexts.” (p. 25)

(4)

“Did children’s choices vary across trials?”

We ran an additional analysis to address this question, indicating no evidence for such an effect:

“Further analysis also confirmed stable risk preferences throughout the test phase (see Supplementary Materials).” (p. 18)

“To test whether children’s risk preferences varied across trials, we fitted a simple monotonic model in brms with the predictor “trial” on children’s risk preferences. This analysis did not reveal any evidence for such an effect (mo_{Trial} (95%-CI) = -0.01 (-0.04; 0.02)). Estimated trial effects are plotted below together with mean probabilities for risk-seeking behaviors.

Figure S5. Estimated monotonic effect of trial order on risk preferences. Dotted line indicates chance level at prob = 0.5; solid line represents model estimates; grey areas indicate 95%-HPD intervals; dots represent mean probabilities per trial” (Supplementary Materials S5)

(5)

“Do we know if the behavior of the children who acted as the peer observers varied significantly from those who did not?”

We offered all children to act as peer observers in the study except the last boys and girls tested at each site – because we did not run the study with additional children here. The only other reason for children not to participate in this role was organizational, such as when children had to leave the study area for lunch or other activities. Thus, the majority of children ($n = 96$) acted as peer observers once in the study, whereas few children engaged in this role twice ($n = 18$) or three times ($n = 4$). Only $n = 26$ children did not participate as observer peers. We are convinced that a representative and substantial subset of children participated as peers, introducing no bias to the current study. We would be happy to conduct additional analyses if you deem such information necessary to further substantiate this claim.

(6)

“The introduction could benefit from a slight reorganization (perhaps adding subheadings?) to make the various components of this investigation more clear: so (1) social context, (2) collaboration, (3) the effect of culture / WEIRD vs. non-WEIRD, and (4) development. And I do think each of those, especially the collaboration and social context sections, can be rounded out with more citations such as: a. Warneken, F., Lohse, K., Melis, A. P., & Tomasello, M. (2011). Young children share the spoils after collaboration. *Psychological science*, 22(2), 267-273. b. Koomen, R., Grueneisen, S., & Herrmann, E. (2020). Children delay gratification for cooperative ends. *Psychological science*, 31(2), 139-148.”

Thank you for these helpful suggestions. As requested, we included sub-headers to the introduction and have re-organized this section for more clarity. We also included the references by Warneken et al. and Koomen et al. to our discussion on collaboration effects on children's behaviors.

(7)

“A few other citations to consider:

- a. Reiter, A. M., Suzuki, S., O'Doherty, J. P., Li, S. C., & Eppinger, B. (2019). Risk contagion by peers affects learning and decision-making in adolescents. *Journal of Experimental Psychology: General*, 148(9), 1494.**
- b. Harvey, T., & Blake, P. R. (2022). Developmental risk sensitivity theory: the effects of socio-economic status on children's risky gain and loss decisions. *Proceedings of the Royal Society B*, 289(1983), 20220712.**
- c. Rohde, I. M., & Rohde, K. I. (2011). Risk attitudes in a social context. *Journal of Risk and Uncertainty*, 43(3), 205-225.**
- d. Defoe, I. N., Semon Dubas, J., & Romer, D. (2019). Heightened adolescent risk-taking? Insights from lab studies on age differences in decision-making. *Policy Insights from the Behavioral and Brain Sciences*, 6(1), 56-63.”**

Thank you for providing these references—we found all of them highly relevant and added them, together with other recent publications, to the manuscript.

(8)

“I also think the argument in the introduction can be strengthened a bit; it could be beneficial to set the argument up in a way that outlines more clearly what’s missing in the literature, e.g. “the majority of studies examining risk preferences consider an individual decision-maker acting on their own; however, we know that in the real world, decisions often take place in social contexts, and there are additional signaling and reputational functions that shape our risky choices”.”

Following this suggestion, we now provide a more concise rationale for the current study.

“While most empirical studies focus on risk preferences of sole decision-makers, real-world economic choices often occur in social contexts, where reputational concerns and interdependence may profoundly impact decision-making. Despite the potent impact of social contexts on human risk preferences, surprisingly little is known about how such effects are shaped developmentally and how they vary cross-culturally.” (p.6)

“The current study investigated the development of children’s risk preference in social context. By manipulating the presence of and interdependence with peers in children's economic decision-making, we explored how age, sex, and cultural background affect how social interactions shape their risk-taking behavior.” (p.7)

(9)

“A minor note, but contrary to lines 112-113, in the Amir 2019 paper (citation 40), we did include gender as a covariate in all of our statistical analyses (it’s reported in our model outputs), and it was never a significant predictor.”

We are sorry that this wording was ambiguous. We did not intend to state that gender was not included in the analyses of this study, but that it was included as a fixed effect not varying across cultures. We clarified the statement accordingly.

“While this research did not report cultural differences in sex disparities for children's risk preferences, it is important to note that the study design did not directly test for such interactions. Instead, sex was modeled as a fixed effect, invariant across cultures.” (p.5)

(10)

“A minor clarification but “we expected younger children to exhibit stronger risk preferences than older participants” (line 213-14) doesn’t specify the order of the prediction. I would update to something like “more risk-tolerance” or “more risk-seeking”, or even “preferences for risk” instead of “risk preference”.”

Agreed. The sentence now reads as follows.

“Following previous work, we expected younger children to be more risk-seeking than older participants.” (p. 9)

Reviewer 2:

(1)

“The paper is well written and organized except for very minor omission of closing bracketed content in a few places which can easily be corrected by the author by editing the paper.”

Thank you for this positive evaluation and the hint to this omission. We have corrected it.

Reviewer 3:

(1)

“However, it is not entirely clear what the scope of these results of comparing these to groups of kids really is, in terms of understanding the development of risk preferences or cultural differences.

In the introduction it becomes clear that risk preference vary between ages, sexes, cultures etc. But in the final cultural part specifically no clear framework or predictions emerge. So in the end we also do not know if these results reject or inform a theory about how culture impacts risk preference development. Of course the resulting difference between the two groups are also basically zero, but whether or not that is an issue for an existing theory I would not know. As it currently is I would therefore not see how it would fit the current journal.”

Thank you for this note. We agree that the current research does not build a new theory on risk preferences, but rather explores the interplay of variables that have been considered relevant in this domain. We have streamlined our manuscript to focus more on the relevance of social and cultural variation in risk preferences for current research and theory. Doing so, we have also strengthened the rationale for comparing children from the two communities in more detail and provide more information on the scope of the study in the discussion. Here, we outline how these results inform current research on risk preferences in childhood, emphasizing the importance of variation across cultures and social contexts. We further contextualize the research within extant literature on the cultural communities hosting this research (p.22, p. 24). Data from rural, subsistence-based communities of the Global South, and Majority World communities more generally, are vastly underrepresented in mainstream research in psychology and economics. This is why we are optimistic that this research, albeit largely exploratory, can inform scientific debates about the developmental origins of risk preferences in social and asocial contexts.

“Furthermore, prior research studying social context effects on risk preferences almost exclusively sampled urban, industrialized, and wealthy communities in the Global North, seriously limiting the generalizability of such research beyond urban, industrialized, and wealthy communities. A relevant dimension in this regard concerns societal emphases on psychological autonomy compared to hierarchical relatedness (Keller, 2012, 2016). Societies in which individuals are equipped with high levels of psychological autonomy, navigating social interactions emphasizing their subjective desires and interests, may be less susceptible to social contexts as decisions can be made without strong social obligations. Given the role of autonomy as a foundational schema among many hunter-gatherer societies (Barnard, 2002; Hewlett, 2016), studying peer presence effects on risk preferences among hunter-gatherer communities offers a critical test of the robustness of such effects. In contrast, individuals from societies emphasizing hierarchical relatedness may be more susceptible to social context when navigating economic risks, given their dense networks of social relations and adherence to social norms (Keller & Kärtner, 2013).

To gain a richer understanding of how social environments shape children's risk preferences, research must encompass cultural communities exhibiting variation in subsistence, but also in socialization goals regarding child autonomy and interdependence. Such perspectives are critical for improving the generalizability and validity of developmental research across psychology and economics.” (p.7)

“To address our cross-cultural research question and adjust the research to the cultural communities we studied, we developed a new risk preference assessment paradigm. This paradigm incorporates core elements from established risk assessments, such as a forced-choice format with safe and risky options of equal expected values, ensuring face validity (Defoe et al., 2015). Crucially, it also allows us to investigate children's risk preferences in diverse social settings, examining effects of peer influence and collaboration under controlled conditions. To promote cultural fairness, we used a haptic device to assess children’s risk preferences, rather than relying on instructed rules or electronic devices that might induce cultural barriers to the assessment. A key challenge of cross-cultural studies lies in balancing standardized procedures with the unique needs of communities outside the market-integrated and technology-affluent Global North. The present study lays a valuable foundation for future research, enabling comparisons between our approach and other established risk preference paradigms.” (p.26)

(2)

“This is also related to a lack of information about the generalizability of the current risk task. How well does it relate to other behavioral tasks? Risk preferences are dependent on so many factors, probability, the presence of a safe choice etc. So it is also hard to compare these results to other studies. (ps. see Defoe et al., for good meta-analyses on developmental patterns, they could be a nice addition for a more updated reference list).”

Thank you for this. We have added a reference to the study by Defoe and colleagues to the revised manuscript. Also, we added a note on the generalizability of the current task to previous work, emphasizing both similarities and novel aspects of the current task that were designed to test the questions at hand. We admit that applying paradigms designed and validated in urban communities of the Global North to rural, subsistence-based communities is challenging.

“To address our cross-cultural research question and adjust the research to the cultural communities we studied, we developed a new risk preference assessment paradigm. This paradigm incorporates core elements from established risk assessments, such as a forced-choice format with safe and risky options of equal expected values, ensuring face validity (Defoe et al., 2015). Crucially, it also allows us to investigate children's risk preferences in diverse social settings, examining effects of peer influence and collaboration under controlled conditions. To promote cultural fairness, we used a haptic device to assess children’s risk preferences, rather than relying on instructed rules or electronic devices that might induce cultural barriers to the assessment. A key challenge of cross-cultural studies lies in balancing standardized procedures with the unique needs of communities outside the market-integrated

and technology-affluent Global North. The present study lays a valuable foundation for future research, enabling comparisons between our approach and other established risk preference paradigms.” (p.26)

(3)

“In sum, it is currently hard to see how these results fit into the larger scheme of things, in relation to both development of risk preferences and cultural comparisons.”

We hope that the revised manuscript better describes how the current results relate to prior research on the cultural and developmental specifics of children’s risk preferences, providing much needed evidence from the Majority World to the study of risk preferences in psychology and economics.

“While most empirical studies focus on risk preferences of sole decision-makers, real-world economic choices often occur in social contexts, where reputational concerns and interdependence may profoundly impact decision-making. Despite the potent impact of social contexts on human risk preferences, surprisingly little is known about how such effects are shaped developmentally and how they vary cross-culturally.” (p.6)

“To gain a richer understanding of how social environments shape children's risk preferences, research must encompass cultural communities exhibiting variation in subsistence, but also in socialization goals regarding child autonomy and interdependence. Such perspectives are critical for improving the generalizability and validity of developmental research across psychology and economics. [...] The current study investigated the development of children’s risk preference in social context. By manipulating the presence of and interdependence with peers in children's economic decision-making, we explored how age, sex, and cultural background affect how social interactions shape their risk-taking behavior.” (p.7)

(4)

“minor. I wondered if it would be interesting to look at order effects.. maybe all risks were taken in the first block? or later when a bunch of safe rewards were harvested? Also it would be good to explain how the left/right bias was still a problem after removing so many subjects. And how you are really sure that a strong left or right bias represent misunderstanding the task and not reflects a real preference.”

Thank you for raising these points (see also similar points by Reviewer 1). We addressed this point in the supplementary materials, finding no evidence for order effects throughout the study-

“Further analysis also confirmed stable risk preferences throughout the test phase (see Supplementary Materials).” (p. 17)

“To test whether children’s risk preferences varied across trials, we fitted a simple monotonic model in brms with the predictor “trial” on children’s risk preferences. This analysis does not reveal any evidence for such an effect (mo_{Trial} (95%-CI) = -0.01 (-0.04; 0.02)). Estimated trial effects are plotted below together with mean probabilities for risk-seeking behaviors.

Figure S5. Estimated monotonic effect of trial order on risk preferences. Dotted line indicates chance level at $prob = 0.5$; solid line represents model estimates; grey areas indicate 95%-HPD intervals; dots represent mean probabilities per trial” (Supplementary Materials S5)

5th Apr 24

Dear Dr Stengelin,

Your manuscript titled "Peer Presence Effects on Risk Preferences in Childhood: Variation Across Development, Sexes, and Cultures" has now been seen by our reviewers, whose comments appear below. In light of their advice I am delighted to say that we are happy, in principle, to publish a suitably revised version in *Communications Psychology* under the open access CC BY license (Creative Commons Attribution v4.0 International License).

We therefore invite you to revise your paper one last time to address the remaining concerns of our reviewers and a list of editorial requests. At the same time we ask that you edit your manuscript to comply with our format requirements and to maximise the accessibility and therefore the impact of your work.

EDITORIAL REQUESTS:

Please make sure to review our policy on Ethics and Inclusion (<https://www.nature.com/commpsychol/editorial-policies/authorship#authorship-inclusion-and-ethics-in-global-research>) and include a statement addressing how local researchers were included in your study.

SUBMISSION INFORMATION:

In order to accept your paper, we require the files listed at the end of the Editorial Requests Table; the list of required files is also available at <https://www.nature.com/documents/commjsj-file-checklist.pdf>.

OPEN ACCESS:

Communications Psychology is a fully open access journal. Articles are made freely accessible on publication under a CC BY license (Creative Commons Attribution 4.0 International License). This license allows maximum dissemination and re-use of open access materials and is preferred by many research funding bodies.

For further information about article processing charges, open access funding, and advice and support from Nature Research, please visit <https://www.nature.com/commspsychol/article-processing-charges>

At acceptance, you will be provided with instructions for completing this CC BY license on behalf of all authors. This grants us the necessary permissions to publish your paper. Additionally, you will be asked to declare that all required third party permissions have been obtained, and to provide billing information in order to pay the article-processing charge (APC).

* **DATA AVAILABILITY:**

[link redacted]

Best regards,

Jennifer Bellingtier

Jennifer Bellingtier, PhD

Senior Editor

Communications Psychology

REVIEWERS' EXPERTISE:

Reviewer #1 Cross-cultural Psychology, Developmental Psychology

Reviewer #3 Developmental Psychology, Decision Making

REVIEWERS' COMMENTS:

Reviewer #1 (Remarks to the Author):

I commend the authors for their revisions, which I believe have adequately addressed my comments.

For the sake of posterity, I would still suggest (in line with Comment #5) that additional analyses be included in the Supplement to address the question of children who acted as peer observers in addition to participating. In particular, (1) the difference between those who did both roles versus those that didn't (which I don't imagine will affect results at all, but would be helpful to have), and (2) whether they participated before observing or vice versa? That part is still a bit unclear to me, the order of the two roles and whether that mattered.

Otherwise, all set on my end.

Dorsa Amir

UC Berkeley

Reviewer #3 (Remarks to the Author):

The authors did address the concerns raised earlier partly. The thrust of the argument seems to be that my concerns are valid, but should be waived because it is interesting but exploratory work, given that risk preferences have not been studied a lot cross-cultures but focus is on Global North. Also the validity remains and here I am less convinced. There are several non verbal risk task for children, like the cups task. And there is quite a literature on how any of these tasks actually correlates with real world risk taking. Given the absence of a framework for understanding or predicting cultural differences in risk preferences between these two groups, the exploratory nature is still not adding to theory building. All in all, I understand that this has been an enormous effort and it does yield interesting data, but still the impact and contribution may be rather limited.

Reviewer 1:

(1)

“For the sake of posterity, I would still suggest (in line with Comment #5) that additional analyses be included in the Supplement to address the question of children who acted as peer observers in addition to participating. In particular, (1) the difference between those who did both roles versus those that didn't (which I don't imagine will affect results at all, but would be helpful to have), and (2) whether they participated before observing or vice versa? That part is still a bit unclear to me, the order of the two roles and whether that mattered.”

We have addressed these questions in our revision.

- (1) Children engaging in both roles did not differ from their peers engaging in one role only.
- (2) Children participating as target child before participating as stooges did not differ from those engaging as stooges first.

As suggested, these questions are addressed in the Supplementary Materials.

“Finally, we tested whether children who participated only as target children ($n = 26$; $n_{\text{Hailom}} = 13$, 6 girls; $n_{\text{Ovambo}} = 13$, 4 girls) showed different risk preferences from their peers who participated in both roles at some point. Note that all children were asked to participate as stooges if they were available at the study location following their participation. We fitted a simple Bayesian regression model with Bernoulli response distribution, predicting children's risk preferences with a dichotomous variable (“One Task Only”). This analysis does not reveal any evidence for such an effect (Estimate (95%-CI) = -0.09 (-0.36 ; 0.18); $\text{probability}_{\text{Target Only}} = .34$, $\text{probability}_{\text{Both Roles}} = .36$).

Also, we investigated whether children who participated as stooges first ($n = 7$; $n_{\text{Hailom}} = 4$, 3 girls; $n_{\text{Ovambo}} = 3$, 1 girl) differed from their peers who participated as target children first. Note that the former group of children participated early in the study phases at each study location to get data collection started. We fitted a Bayesian regression model with Bernoulli response distribution, predicting risk preferences with a dichotomous variable (“Stooge as First Role”). This analysis does not reveal evidence for such an effect (Estimate (95%-CI) = -0.27 (-0.82 ; 0.24); $\text{probability}_{\text{Stooge First}} = .31$, $\text{probability}_{\text{Stooge Second}} = .36$). ”
(Supplementary Materials S6)